# Self-Assembly of Amphiphilic Linear−Dendritic Carbosilane Block Surfactant for Waterborne Polyurethane Coating

**DOI:** 10.3390/polym12061318

**Published:** 2020-06-09

**Authors:** Ruitao Wang, Chunxiang Li, Zhaohua Jiang, Zhijiang Wang

**Affiliations:** School of Chemistry and Chemical Engineering, Harbin Institute of Technology, Harbin 150001, China; wruitao@126.com (R.W.); wangzhijiang@hit.edu.cn (Z.W.)

**Keywords:** linear−dendritic, self-assembly, polyurethane, waterborne coatings, FT-IR

## Abstract

The traditional two-component waterborne polyurethane coating system cannot effectively inhibit the undesirable side reaction between polyisocyanate and water during curing hardening. It is difficult to avoid the microbubbles formed by this reaction during the film formation process, which severely degrades the appearance and decreases the performance of the film. Therefore, the addition of an amphiphilic Linear-Dendritic carbosilane Block Surfactant (LDBS) to the hardener can physically separate the polyisocyanate emulsion from water through self-assembly. The bubble-free film thickness (BFFT) of the two-component waterborne polyurethane coating in this study is approximately 1.5-fold greater than commercial waterborne polyurethane coatings in today’s coating industry. Fourier transform infrared spectroscopy (FT-IR) varied the effectiveness of LDBS for inhibition of the undesirable side reaction. The successful application of the waterborne polyurethane coating with LDBS on the 600 km/h high-speed maglev train provides a technical solution for large-scale industrialization of waterborne polyurethane coating and complete replacement of solvent polyurethane coating.

## 1. Introduction

Polyurethane coating is an important top-coat for agriculture, construction equipment and rail transit vehicles in the paint industry because of its excellent decoration and weather-resistant capability [1,2,3]. However, traditional solvent-based polyurethane coatings are changing to more environment-friendly waterborne coatings due to environmental pollution concerns [4,5]. The two-component waterborne polyurethane coatings are generally composed of a base and hardener [6]. The main components of the base are hydroxyl polyacrylate dispersion, pigments, additives and water. In the case of the hardener, it mainly consists of polyisocyanate resin, which is dissolved in a co-solvent [7,8]. The base and hardener are usually mixed during the application process, whereas polyisocyanate is dispersed in water to form the emulsion [9,10]. After the evaporation of water, polyisocyanate reacts with the hydroxyl resin to form the polymer and solidify into film. The basic reactions are given by Equations (1)–(4) [11,12].


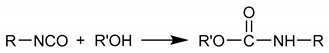
(1)

Since polyisocyanate is very active, some side reactions take place during the mixing of base and hardener (see Equations (2)–(4)).


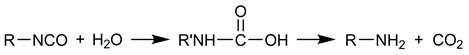
(2)


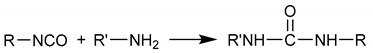
(3)


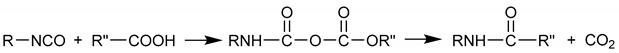
(4)

It is worth noting that the side reaction between polyisocyanate and water, originating from the base and moisture, produces carbon dioxide gas [13]. This will cause defects, because if the applied wet film is too thick, the carbon dioxide gas, which cannot escape, will cause bubbles to form. In the polyurethane coatings, bubble-free film thickness (BFFT) is used to describe the index of two-component waterborne polyurethane coatings [14]. At present, BFFT of waterborne polyurethane coatings is about 80 μm, which is lower than that of solvent-based coatings (which is about 100 μm) [15].

For example, rail transit vehicles use high-speed electric multiple unit (EMU) coatings, which need to provide protection to the vehicle’s body for a long period of time due to abrasion caused by sand, dust, air, etc. [16,17]. Therefore, the standard coating thickness of polyurethane for high-speed EMU needs to reach 60–80 μm, while in some areas, because of repeated spraying, it reaches values >1.5 times that of the standard film thickness, which increases the dry film thickness to more than 100 μm [18,19]. The thickness of the coatings, which can be applied simultaneously, is limited by blistering defects. Hence, low BFFT is a problem that needs to be resolved in large-scale applications of waterborne polyurethane coatings [20].

There are two common technical methods for improving the BFFT of waterborne polyurethane coatings. (1) Hydrophobic low-viscosity polyisocyanate (such as Desmodur N 3900 of Covestro), which inhibits the side reaction between polyisocyanate and water. Although BFFT can be effectively increased to about 100 μm, low-hydrophilicity polyisocyanates are prone to poor dispersion in water. In the absence of the mechanical disperser, the size of the polyisocyanate emulsion becomes too large and chemical crosslinking with the hydroxyl polyacrylate resin is insufficient, resulting in poor performance [21,22,23]. (2) Prolonged opening time by adding excessive co-solvents allows the produced carbon dioxide gas sufficient time to escape. The disadvantage of this method is that the contents of volatile organic compounds (VOCs) are too high, which affects the environmental friendliness of waterborne polyurethane coatings [24,25].

This study examines the effect of the addition of amphiphilic linear—dendritic carbosilane block surfactant (LDBS) on BFFT’s improvement of two-component waterborne polyurethane coating. Initially, LDBS was added to the hardener (polyisocyanate solution). When the polyisocyanate was emulsified in the water of the base, LDBS self-assembled into a hydrophobic interface to physically isolate water and isocyanate. This approach inhibits the side reaction, which produces carbon dioxide gas, reduces bubble defects and improves BFFT. Furthermore, the BFFT of the waterborne polyurethane coating was evaluated based on its appearance after drying in standard atmosphere. The appearance of the coatings was investigated using optical microscopy. In addition, the inhibitory effects of LDBS on the side reaction were evaluated using FT-IR.

## 2. Experimental

### 2.1. Materials and Instruments

The chemicals used in the current study included monomethoxy polyethylene glycol (molecular weight of 750 g/mol; Sigma-Aldrich (Shanghai, China) Trading Co., Ltd), allyl bromide (analytical grade reagent; mass fraction of 98%; Aladdin Reagent (Shanghai, China) Co., Ltd.), sodium hydride (analytical grade reagent; mass fraction of 60%, dispersed in mineral oil, Aladdin Reagent (Shanghai, China) Co., Ltd.), trichlorosilane (analytical grade reagent; mass fraction of 99%; Aladdin Reagent (Shanghai, China) Co., Ltd.), Karstedt catalyst (analytical grade reagent; w(Pt) = 2% dimethylbenzene solution; Aladdin Reagent (Shanghai, China) Co., Ltd.), tetrabutylammonium iodide (analytical grade reagent; mass fraction of greater than 99.0%; Aladdin Reagent (Shanghai, China) Co., Ltd.), magnesium powder (analytical grade reagent; mass fraction of 99.5%; Aladdin Reagent (Shanghai, China) Co., Ltd.), triethylsilane (analytical grade reagent; mass fraction of 98%; Energy Chemical Technologies (Shanghai, China) Co., Ltd.), Macrynal VSM 6299 w/42 WA (hydroxyl polyacrylate dispersion; Allnex Resins (Suzhou, China) Co., Ltd.), Kronos 2160 (pigment; Leverkusen, Germany Kronos International Inc.), Surfynol 104 BC (wetting agent; Evonik Specialty Chemicals (Shanghai, China) Co., Ltd.), Additol VXW 6208 (dispersing agent; Allnex Resins (Suzhou, China) Co., Ltd.), Borchigel 0620 (thickener; Borchers (Shanghai, China) Trading Co., Ltd.), Additol VXW 5929 (leveling agent; Allnex Resins (Suzhou, China) Co., Ltd.), BYK 011 (defoamer; BYK Additives (Shanghai, China) Co.Ltd.), Texanol (solvent; Eastman (Shanghai, China) Chemical Commercial Co., Ltd.), Bayhydur XP 2655 (curing agent; Covestro Polymers (Shanghai, China) Co., Ltd.), and Dowanol PGDA (solvent; Dow Chemical (Shanghai, China) Co., Ltd.).

After the coating was dried and cured, the dry film thickness was measured by Elcometer 456 coating thickness gauge (Elcometer Limited, Manchester, UK) and the appearance of the coating was observed using 200× optical microscope (Keyence Corporation, Shanghai, China) to evaluate BFFT. The chemical difference in each film with different amounts of LDBS in coatings was studied using attenuated total reflectance Fourier transform infrared spectroscopy (ATR FT-IR) on a Thermo Fisher Nicolet iS10 instrument (Thermo Fisher Scientific, Shanghai, China) equipped with a mercury-cadmium-telluride detector (MCT) cooled with liquid nitrogen and OMNIC software at a spectral resolution of 4 cm^−1^ and wavenumber range of 4000−650 cm^−1^. This process was repeated in triplicate. The quantitative analysis of characteristic functional groups was carried out according to Beer-Lambert law, whereas attention was paid to carbamate carbonyl and urea carbonyl groups in order to analyze the inhibitory effect of LDBS on side reactions, whose product was polyurea.

### 2.2. Preparation of LDBS

The synthesis, structural characterization and application of LDBS are important areas of research; however, in this study we will focus on the application of LDBS. The preparation of LDBS was mainly achieved according to the mature research work of Youngkyu Chang and Wang Xi [26,27]. In brief, first, the terminal hydroxyl groups of mPEG750 were allylated at room temperature for 24 h. Then, repeated hydrosilylation and allylation were carried out at room temperature for 1 h and then at 45 °C for 10 h. The organosilicon groups were connected to the end of linear polyethylene glycol in a divergent manner. Finally, PEO-Si-2G silicone-carbon block copolymers were obtained, as shown by Scheme 1.

The physical and chemical parameters of PEO-Si-1G and PEO-Si-2G are listed in Table 1. It can be seen that PEO-Si-2G contains 49.5% weight of the hydrophobic dendrimer block. Compared to 19.3% of PEO-Si-1G, PEO-Si-2G tends to form a physical isolation interface with water. Furthermore, PEO-Si-2G has nine end groups, which means it has a more dendrite structure, making it easier to spread at the interface of polyisocyanate droplets [26].

### 2.3. Preparation of Waterborne Polyurethane Coatings

The waterborne polyurethane coating is composed of two parts: base and hardener. The formulation and raw materials of the base and hardener are listed in Table 2 and Table 3.

The components were added consecutively with stirring at 25 °C, homogenized for 20 min, and de-gassed for 24 h.

Both the base and hardener of waterborne polyurethane coatings were mixed in 5:1 ratio (wt./wt.). The system was stirred mechanically for 3 min, diluted with deionized water to 20%, and mixed evenly again. The coatings were sprayed on a tinplate panel using SATA jet 5000−110 air spray gun. The coating was applied under the standard conditions of 25 °C and 50 relative humidity (RH). The evaluations and tests were carried out after 7 days after drying at 25 °C and 50 RH.

## 3. Results and Discussion

### 3.1. Mechanisms of LDBS Self-Assembly to Form Physically Isolated Interface

In this study, LDBS was added to the curing agent, allowing the resin particles to form a physical separation between the polyisocyanate emulsion and water, which effectively reduces the side reaction of polyisocyanate and water. Hence, this promotes reduction of carbon dioxide gas production, enhancement of BFFT of two-component waterborne polyurethane coatings in the film forming process, and improvement of the application performance and film appearance.

LDBS is a kind of specially designed amphiphilic surfactant, in which the hydrophilic group is linear and the lipophilic group is branched. After the addition of the polyisocyanate curing agent to LDBS and mixed into the main agent to form the emulsion, the branched lipophilic group self-assembles under hydrophobic weak force and spreads on the surface of the isocyanate droplet to form the physical separation between water and isocyanate [28]. This is due to inhibition of the side reaction, as shown in Figure 1. The linear hydrophilic group is perpendicular to the branched lipophilic group network, which enhances the spreading of the lipophilic network plane on the isocyanate surface and stability of the morphology.

### 3.2. Evaluation of the BFFT

The water-based polyurethane was prepared according to the formulation presented in Table 1. Six different amounts of LDBS lying within the range of 0.25–1.5% in hardener and blank samples were selected in order to compare the impact on BFFT (Table 4). The results show that the addition of LDBS improves BFFT. With the addition of only 0.25% LDBS, the BFFT increases significantly from 80.4 μm for sample without LDBS to 92.1 μm. Additionally, BFFT increases with increasing LDBS. As the amount of LDBS added increases from 0.75% to 1.25%, BFFT reaches its maximum value of 120 μm, which is equivalent to around 50% increase compared to the initial BFFT value.

The amount of LDBS added and influence of LDBS are shown in Figure 2. It can be clearly seen that BFFT increases with increasing amount of LDBS. When the amount of LDBS is around 1.0% the BFFT value begins to show a downward trend.

To accurately find the optimum dosage of LDBS, five different amounts were selected within the range of 0.75–1.25% (Table 5). It was found that the addition of 1.0% LDBS is the optimal amount for the studied system.

Therefore, 1.0% LDBS is suitable to achieve fully coated polyisocyanate droplets, which can well block the physical contact between water and polyisocyanate droplets, thus reducing the side reaction between water and isocyanate groups. Below 1.0% LDBS, insufficient blocking is observed, but above this value is also unfavorable due to the foam’s stabilizing effect on the surfactant.

Figure 3 shows the comparison between the appearances of the film prepared by 1.0% LDBS addition and the blank coating sample without LDBS. The film with a thickness of 120.8 μm (Figure 3a) shows the coating after the addition of 1.0% LDBS, while the film with a thickness of 120.6 μm (Figure 3b) shows the coating without LDBS. The thicknesses of the two samples are comparable. However, examination of the film’s appearances under fluorescent tube light, it can be seen that there are obvious micro-bubbles on the surface of the sample without LDBS, while the sample with 1.0% LDBS coating has a good appearance and no coating defects.

The above samples were observed using 200× optical microscope, as shown in Figure 4a,b. Figure 4b shows the magnified appearance without the addition of LDBS, where the bubbles on the surface are clearly observed, while the coating surface with 1.0% LDBS appears smooth (Figure 4a). Through visual observation, LDBS has an obvious effect on improving BFFT of the coating.

### 3.3. FT-IR Spectroscopy Analysis

Attenuated total reflection Fourier transform infrared spectroscopy (ATR FT-IR) was used to analyze the coating films depicted in Figure 4a,b. The contrast infrared spectra are shown in Figure 5.

The generation of a carbonyl group was the main outcome of the polyurethane curing reaction. The types of carbonyl groups formed by the main and side reactions were different. In waterborne polyurethane coating, two main forms of carbonyl groups were detected; namely the urethane and urea carbonyl group. Among them, carbamate carbonyl corresponded to the main product (Equation (1)), while urea carbonyl corresponded to the side products (Equations (2) and (3)), which would produce carbon dioxide gas.

As shown in Figure 5, the peak between 1780–1620 cm^−1^ represents the characteristic peak area of the carbonyl group [29]. The absorption peak of carbamate carbonyl is found between 1702–1740 cm^−1^, and that of urea carbonyl between 1630–1689 cm^−1^. Since the carbamate and urea carbon are the main and side products, respectively, in the curing process of waterborne polyurethane coating, the influence of LDBS on the inhibition of side reaction can be analyzed.

As shown in Figure 5, the spectrum of the coating film of waterborne polyurethane coating without LDBS (red spectrum) displays a peak at 1716.38 cm^−1^, which corresponds to the urethane carbon group, whereas that at 1684.58 cm^−1^ relates to the urea carbon group. In the spectrum of the coating film of waterborne polyurethane coating containing 1.0% LDBS (blue spectrum), the peak at 1716.99 cm^−1^ corresponds to the urethane carbon group and that at 1684.63 cm^−1^ relates to the urea carbon group.

Through Beer-Lambert law, a quantitative comparison between the carbamate carbonyl and urea carbonyl was carried out (Equation (5)).
(5)$$X={\left(1+\frac{aoAn}{anAo}\right)}^{-1}$$
where “X” is the fraction (%) of the carbamate carbonyl functional group in the sum of carbamate carbonyl and urea carbonyl functional groups, “A” is the absorbance, “a” is the molar absorption coefficient, and the subscripts “*o*” and “*n*” are the carbamate carbonyl and urea carbonyl, respectively.

The main reaction ratio can be calculated according to the area of carbonyl absorption peak of carbamate and the area of the urea carbonyl absorption peak in the infrared spectrum using Equation (6).
(6)$$X=\frac{\sum So}{\sum \left(So+Sn\right)}$$
where “So” is the peak area of carbamate carbonyl absorption, and “Sn” is the peak area of urea carbonyl absorption.

The area of the characteristic peaks of carbamate and urea carbon was integrated (Figure 5). The integration process is shown in Figure 6, Figure 7, Figure 8 and Figure 9.

Using these integrated area data, the ratio of the main product was calculated according to Lambert-Beer law [30,31]. The calculated results are presented in Table 6.

According to the results presented in Table 6, when 1.0% LDBS is added to the curing agent it promotes a significant increase in the proportion of the main product from 35.38% of the blank sample to 43.67%. Hence, the addition of LDBS effectively reduces the proportion of the side products, which in turn reduces carbon dioxide gas production and micro-bubble formation in the coating. It is shown that LDBS can inhibit the occurrence of side reactions and improve the effectiveness of BFFT.

### 3.4. Industrial Application

Based on this study, some suggestions have been made for the industrial application of the proposed coating systems. On the exterior surface of the 600 km/h high-speed maglev train of CRRC, the waterborne polyurethane finish with 1.0% LDBS was used.

When the high-speed maglev train runs at the speed of 600 km/h, abrasion of the coating due to sand particles, rain and even air cannot be ignored. Therefore, considering the wear allowance and coating life when designing the coating system, the polyurethane finish needs to be more than 200 μm in thickness. If the traditional waterborne polyurethane coating is used, it must be applied by spray at least three times. Based upon the results of this study, it is feasible to reduce one spraying operation, and also produce a more compact coating without microbubbles. 

The coating showed good application performance and apparent decoration effect. In practical application, when the dry film thickness of waterborne polyurethane finish is above 100 μm for one spray, no micro-bubbles are observed on the surface of the coating by 200× optical microscope. The effect after application is shown in Figure 10.

## 4. Conclusions

In two-component waterborne polyurethane coatings, amphiphilic linear-dendritic carbosilane block surfactant can be dispersed through self-assembly on polyisocyanate emulsion particles. The hydrophobic three-dimensional structure can effectively separate polyisocyanate from water and thus reduce the occurrence of side reactions, which produces carbon dioxide gas. In this way, the micro-bubbles caused by the escaping carbon dioxide gas can be reduced. The bubble free film thickness (BFFT) can be increased, and the appearance of the film can be improved. The results show that the BFFT can be effectively improved with the addition of LDBS to the curing agent of the two-component waterborne polyurethane coating. When the amount of LDBS is 1.0%, BFFT of the waterborne polyurethane coating can be increased to about 120 μm, which is about 1.5 times of the current technical level. The successful application in high speed maglev vehicle of 600 km/h speed provides an industrial solution for waterborne polyurethane coating.

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
