# Peer review of "Self-Assembly of Amphiphilic Linear−Dendritic Carbosilane Block Surfactant for Waterborne Polyurethane Coating"

_polymers, 2020, doi:10.3390/polym12061318_

Round 1
Reviewer 1 Report
The article present the work about the preparation of two-component waterborne polyurethane coatings as well as their suitable in applicability, denoting the value of the products. Nevertheless, I think that the systems should be deeply discussed. Therefore, I recommend the work for its publication after major revisions:
In general, I recommend to authors to revise the English language of the manuscript. As well, I suggest to check and homogenize the Figure, Table and Scheme caption according to the Journal guide. For example, Scheme 2 caption appeared twice, and Figure 4 caption is divided in 2 Figures, please revise it.
In the introduction, (line 29) authors state that “The waterborne polyurethane coatings are generally composed of a base and a hardener”. It is true that it is a versatile and broad area in the field of waterborne polyurethane coatings, but there are also other type of coatings. I recommend to revise this statement.
In the experimental section, authors referred the synthesis of Linear - Dendritic carbosilane Block Surfactant (LDBS) according to previous works (line 98). Nevertheless, considering the importance of these compounds on the final products, I recommend to include briefly the synthesis conditions in the article.
Authors comment in lines 121-123 that the synthesized PEO-Si-3G cannot be used as amphiphilic surfactant, based also on other reported results. In that case, why have authors synthesized this compound if they already knew about the useless behavior of this structure for this type of systems? Have authors determined trough experimental analysis the behavior of this PEO-Si-3G compounds and that it is not valid for the preparation of these systems?
I recommend to revise the overview of 3.1. section. Table 1 and 2 are referred to the composition of the prepared systems, so I recommend including them in the experimental section (they do not include results). As well, in line 139 and 140 some results are included without comments about them (Moreover, the equipment and specifications of these analysis are not included in the experimental section). The results of table 3 are poorly discussed (references…).
In section 3.3., I recommend to discuss deeply the relationship between LBDS and BFFT results.
Author Response
Dear Reviewer & Editor:
Thank you very much for your kind letter, along with the constructive comments of the reviewer concerning our manuscript (polymers-811086). We have thoroughly considered all the comments of the reviewers and substantially revised our manuscript, and the major revised portions are marked in yellow in our revised manuscript. We also respond point by point to the reviewer’s comments as listed below, along with a clear indication of the location of the revision. We look forward to hearing from you.
Thanks again
Sincerely yours
Zhaohua Jiang
Response to Reviews
Question 1:
In general, I recommend to authors to revise the English language of the manuscript. As well, I suggest to check and homogenize the Figure, Table and Scheme caption according to the Journal guide. For example, Scheme 2 caption appeared twice, and Figure 4 caption is divided in 2 Figures, please revise it.
Answer:
Thank you for your professional advice on language. The revised version has been revised by a professional translation company. At the same time, the typesetting has also been revised, especially the two issues mentioned by the reviewer have been revised.
Question 2:
In the introduction, (line 29) authors state that “The waterborne polyurethane coatings are generally composed of a base and a hardener”. It is true that it is a versatile and broad area in the field of waterborne polyurethane coatings, but there are also other type of coatings. I recommend to revise this statement.
Answer:Thank you for the reviewer's professional suggestions in the field of polyurethane coating. I have modified this expression (page 1: 32-33 line).
Question 3:
In the experimental section, authors referred the synthesis of Linear - Dendritic carbosilane Block Surfactant (LDBS) according to previous works (line 98). Nevertheless, considering the importance of these compounds on the final products, I recommend to include briefly the synthesis conditions in the article.
Answer:We are grateful to the reviewer for his suggestion, I have supplemented the reaction temperature and reaction time information of the synthesis (page 4: 123-125 line).
Question 4:
Authors comment in lines 121-123 that the synthesized PEO-Si-3G cannot be used as amphiphilic surfactant, based also on other reported results. In that case, why have authors synthesized this compound if they already knew about the useless behavior of this structure for this type of systems? Have authors determined trough experimental analysis the behavior of this PEO-Si-3G compounds and that it is not valid for the preparation of these systems?
Answer:Thank the reviewers for their logical suggestions. It is true that the preparation of PEO-Si-3G does not play a role in this study. I have deleted this part in the revised draft.
Question 5:
I recommend to revise the overview of 3.1. section. Table 1 and 2 are referred to the composition of the prepared systems, so I recommend including them in the experimental section (they do not include results). As well, in line 139 and 140 some results are included without comments about them (Moreover, the equipment and specifications of these analysis are not included in the experimental section). The results of table 3 are poorly discussed (references…).
Answer:Thanks for the reviewer's suggestions on the paper structure. I have adjusted part 3.1 to part 2.3 of the second part, so that the article is more organized. Thank the reviewer!
Question 6:
In section 3.3., I recommend to discuss deeply the relationship between LBDS and BFFT results.
Answer:According to the reviewer's suggestion, we add a paragraph to discuss the relationship between LDBS and BFFT (page 7: 204-208 line).
In addition, we carefully checked the English in the text and the mistakes including some grammar and spelling errors again. And all figures and the sequence of the references are renewed in the text.
Reviewer 2 Report
The subject of the paper is very interesting for an application reasons. However, the manuscript needs to be supplemented with the information on materials and methods. I also have reservations about the conducted investigation.
Here are the comments asking for a reference to the authors.
- p.3, l.92: What is the manufacturer of FTIR spectrometer?
- p.3, l.93: What parameters where used for this investigation? What was a wavenumber range? How many scans have been recorded?
- p.3, l.96: What device were used for taking the microscopic images of the sample surfaces? How was the layer thickness measured?
- p.3, l.99: Introduce the shortcut explanation first.
- p.3-4: Please, correct the compound of trichlorosilane and the triethylsilane radicals in Scheme 1 and 2.
- p.3-4: How did you estimate the structure of PEO-Si-nG compounds?
- p.5, l.122: Why did you synthesize PEO-Si-3G, since you expected it to be unsuitable for coating synthesis?
- p.5, l.126: What were the spraying conditions?
- p.5-6, Tables 1 and 2: Why are these materials not included in the "Materials" section? Were they prepared in some way before being added to the reaction mixture?
- p.6, l.138: What were the synthesis conditions? What was the temperature?
- p.6, l.144: Should be PEO-Si-2G.
- p.6, Table 3: Describe a CMC abbreviation.
- p.7, l.174: should be: “….the BFFT increased significantly from 80.4μm for sample without LDBS…”
- p.13: Table 6: According to my knowledge, only the standard peak areas can be compared. Did you do this? Direct comparison of peaks is not reliable.
- p.14: Has the surface of the coating also been assessed after use on this high-speed maglev train?
- p.14: Please enter the references according to the publisher's instructions.
Author Response
Dear Reviewer & Editor:
Thank you very much for your kind letter, along with the constructive comments of the reviewer concerning our manuscript (polymers-811086). We have thoroughly considered all the comments of the reviewers and substantially revised our manuscript, and the major revised portions are marked in yellow in our revised manuscript. We also respond point by point to the reviewer’s comments as listed below, along with a clear indication of the location of the revision. We look forward to hearing from you.
Thanks again
Sincerely yours
Zhaohua Jiang
Response to Reviews
Question 1:
p.3, l.92: What is the manufacturer of FTIR spectrometer?
Answer:Thanks for the reviewer's suggestion, the information of test instrument has been added to the revised draft (page 3: 110 line).
Question 2:
p.3, l.93: What parameters where used for this investigation? What was a wavenumber range? How many scans have been recorded?
Answer:Thanks for the reviewer's suggestion, we have added the details of FTIR test process to the paper according to the reviewer's requirements (page 3: 112-113 line).
Question 3:
p.3, l.96: What device were used for taking the microscopic images of the sample surfaces? How was the layer thickness measured?
Answer:Thanks for the reviewer's suggestion, the information of optical microscope and dry film thickness gauge has been added to the revised draft (page 3: 104-107 line).
Question 4:
p.3, l.99: Introduce the shortcut explanation first.
Answer:Thanks for the reviewer's suggestion! Explanation has been added as required (page 4: 119-122 line).
Question 5:
p.3-4: Please, correct the compound of trichlorosilane and the triethylsilane radicals in Scheme 1 and 2.
Answer:Thanks to the reviewer for his careful review, the relevant content has been changed as required, and SI has been changed to Si (page 4: 132-134 line).
Question 6:p.3-4: How did you estimate the structure of PEO-Si-nG compounds?
Answer:Thanks for the reviewer's suggestion. We have added more explanation on this aspect (page 4: 119-124 line).
Question 7:
p.5, l.122: Why did you synthesize PEO-Si-3G, since you expected it to be unsuitable for coating synthesis?
Answer:Thank the reviewers for their logical suggestions. It is true that the preparation of PEO-Si-3G does not play a role in this study. I have deleted this part in the revised draft.
Question 8:
p.5, l.126: What were the spraying conditions?
Answer:Thanks for the reviewer's suggestion, the information about spraying conditions has been added (page 6: 158-159 line).
Question 9:
p.5-6, Tables 1 and 2: Why are these materials not included in the "Materials" section? Were they prepared in some way before being added to the reaction mixture?
Answer:Thanks for the reviewer's suggestion, we have supplemented the relevant raw material information (page 3: 99-103 line).
Question 10:
p.6, l.138: What were the synthesis conditions? What was the temperature?
Answer:Thanks for the reviewer's suggestion, we have increased the information about synthesis reaction time and temperature (page 4: 123-125 line).
Question 11:
p.6, l.144: Should be PEO-Si-2G.
Answer:Thank you for your careful review. We have revised the relevant content (page 4: 137 line).
Question 12:
p.6, Table 3: Describe a CMC abbreviation.
Answer:Thanks for the reviewer's suggestion, the information about CMC abbreviation has been added (page 5: 144 line).
Question 13:
p.7, l.174: should be: “….the BFFT increased significantly from 80.4μm for sample without LDBS…”
Answer:Thanks for the careful review of the reviewer, and the relevant information has been revised (page 6: 187-188 line).
Question 14:
p.13: Table 6: According to my knowledge, only the standard peak areas can be compared. Did you do this? Direct comparison of peaks is not reliable.
Answer:Thanks for the reviewer's suggestion, the comparison data has been compared with peak area, and the table has been revised to clarify again (page 11: 280 line).
Question 15:
p.14: Has the surface of the coating also been assessed after use on this high-speed maglev train?
Answer:Thanks for the reviewer's suggestion, the relevant information about the observation results of the coating surface of the maglev has been added in the article (page 11: 304-305 line).
Question 16:
p.14: Please enter the references according to the publisher's instructions.
Answer:Thanks for the reviewer's suggestion, the format of the references in the article has been modified according to the requirements of the journal.
In addition, we carefully checked the English in the text and the mistakes including some grammar and spelling errors again. And all figures and the sequence of the references are renewed in the text.
Round 2
Reviewer 1 Report
After the revision of all the comments mentioned by the reviewers.
I consider that the article is acceptable for its publication in the present form.
Reviewer 2 Report
Thank you for considering my comments. I believe that your research results are of great application importance. I wish you success in introducing your coating to use.
Please correct the Equation 5 and 6. The letter markers "o", "n" should be written as subscripts.